# EGFR T751_I759delinsN Mutation in Exon19 Detected by NGS but Not by Real-Time PCR in a Heavily-Treated Patient with NSCLC

**DOI:** 10.3390/ijms232113451

**Published:** 2022-11-03

**Authors:** Zi-Ting Chang, Tien-Ming Chan, Chiao-En Wu

**Affiliations:** 1Department of Internal Medicine, Chang Gung Memorial Hospital at Linkou, College of Medicine, Chang Gung University, Taoyuan 333, Taiwan; 2Division of Rheumatology, Allergy and Immunology, Chang Gung Memorial Hospital at Linkou, College of Medicine, Chang Gung University, Taoyuan 333, Taiwan; 3Division of Hematology-Oncology, Department of Internal Medicine, Chang Gung Memorial Hospital at Linkou, College of Medicine, Chang Gung University, Taoyuan 333, Taiwan

**Keywords:** next-generation sequencing (NGS), non-small cell lung cancer (NSCLC), epidermal growth factor receptor (EGFR), exon 19 deletion, afatinib, osimertinib

## Abstract

The detection of driver gene mutations can determine appropriate treatment strategies for non-small cell lung cancer (NSCLC) by identifying the presence of an effective druggable target. Mutations in the gene encoding the epidermal growth factor receptor (*EGFR*) are common driver mutations in NSCLC that can be effectively targeted by the use of EGFR tyrosine kinase inhibitors (EGFR-TKIs). However, without the detection of driver mutations, appropriate therapeutic decisions cannot be made. The most commonly applied methods for detecting driver gene mutations are assays based on polymerase chain reaction (PCR). However, the underlying mechanism of PCR-based assays limits the detection of rare mutations. Therefore, patients harboring rare mutations may not receive optimal treatment. We report a heavily-treated patient with NSCLC who harbored a T751_I759delinsN mutation in exon 19 of *EGFR* that was not detected by real-time PCR but was successfully detected by next-generation sequencing (NGS). The detection of a driver mutation using NGS resulted in the administration of targeted therapy, leading to favorable progression-free survival for the patient. Our report highlights the importance and potential of routine NGS testing among NSCLC patients for whom traditional assays fail to detect driver mutations when determining treatment options.

## 1. Introduction

The detection of driver gene mutations plays an important role in determining the optimal treatment course for patients with non-small cell lung cancer (NSCLC). Epidermal growth factor receptor (EGFR) is the most common druggable target, and driver mutations in *EGFR* have been identified in 20% of NSCLC patients among Western populations and in 50–60% of patients among Asian populations [1,2]. Therefore, testing for *EGFR* mutations is strongly recommended, as the detection of *EGFR* mutations determines whether EGFR tyrosine kinase inhibitor (EGFR-TKI) therapy is likely to be effective [3].

Methods relying on polymerase chain reaction (PCR) are commonly used in the clinical setting to detect *EGFR* mutations. The Scorpion Amplification Refractory Mutation System (ARMS; QIAGEN therascreen^®^ EGFR; Qiagen, Inc., Valencia, CA, USA) and the cobas^®^ EGFR Mutation Test v2 (Roche Diagnostics, Indianapolis, IN, USA) are two widely used, useful, rapid, and cost-effective companion diagnostic methods for determining the optimal treatment of NSCLC patients. However, these methods fail to detect all *EGFR* mutations, particularly exon 19 deletions [4]. A major limitation of PCR-based methods is the need for appropriately designed primers, which typically do not include all possible exon 19 deletion variations. Reducing the detection of false negatives is therefore crucial to improve the outcomes of clinical testing for *EGFR* mutations and the identification of all patients with NSCLC who harbor *EGFR* mutations and might benefit from EGFR-TKIs.

Gene mutations play a decisive role in predicting the response to EGFR-TKI therapy, requiring highly sensitive mutation detection methods. For patients with rare *EGFR* mutations, next-generation sequencing (NGS) may represent a preferable method to allow for the comprehensive testing of genomic alterations with limited samples [5]. Compared with NGS, the false-negative rate of the cobas^®^ EGFR mutation test was 11.3% [6]. In addition, NGS has been able to identify rare oncogenic *EGFR* mutations, such as the T751_I759delinsS *EGFR* mutation, which was not detected by the Scorpion or ARMS methods [7]. A report demonstrated that NGS was able to confirm all sensitizing mutations detected by matrix-assisted laser desorption/ionization-time of flight (MALDI-TOF) mass spectrometry (MS) and allowed for the characterization of uncommon in-frame complex deletions, and NGS was able to detect the T790M mutation at rates similar to those detected by digital droplet PCR (ddPCR) [8]. We report a patient with lung adenocarcinoma who was found to harbor an exon 19 deletion (T751_I759delinsN) and a T790M mutation in *EGFR,* which were both successfully detected by NGS but not with PCR. Following the detection of these mutations, the patient achieved a good response to EGFR-TKIs.

## 2. Case Report

In 2013, a 67-year-old non-smoking woman was diagnosed with lung adenocarcinoma that metastasized to the liver. Two small cortical metastases involving the right occipital lobe were also noted in 2017. The lung cancer was identified as expressing wild-type *EGFR* according to the PCR-based method (ARMS; QIAGEN therascreen^®^ EGFR; Qiagen, Inc., Valencia, CA, USA), and neither anaplastic lymphoma kinase (ALK) nor proto-oncogene receptor tyrosine kinase (ROS1) alterations were detected by immunohistochemistry (IHC). Because no driver gene mutations were detected, the patient received multiple-line chemotherapy, including the sequential administration of combination cisplatin and pemetrexed, navelbine, and docetaxel.

In June 2020, the patient was diagnosed with progressive brain metastasis, which was treated by the surgical removal of brain tumors. IHC analysis revealed that the brain tumors expressed the lung adenocarcinoma marker thyroid transcription factor (TTF)-1, confirming metastasis from the lung. Genetic testing for *EGFR*/*ALK*/*ROS1* mutations was also performed. The PCR-based testing showed no mutations in exons 18–21 of *EGFR,* and IHC results were negative for both *ALK* and *ROS1* mutations. Because this patient was a woman with no history of smoking, the odds that she developed lung adenocarcinoma with no driver mutations were low. Therefore, we suggested that the patient undergo NGS to investigate the presence of other potential driver mutations that might be associated with targeted therapy. NGS-based comprehensive genetic profiling was performed on a formalin-fixed, paraffin-embedded (FFPE) brain tumor specimen using a targeted panel of 40 actionable genes, 13 fusion genes, and more than 350 fusion transcripts (ACTDrug^®^+, ACT Genomics) able to identify mutations with matched therapeutic options [9]. Oral TS-1 was prescribed from August 2020 to October 2020 while awaiting the NGS test results. The liver and brain metastatic tumors remained stable with TS-1 treatment (Figure 1). A T751_I759delinsN mutation in *EGFR* was detected by NGS (Figure 2). The patient was subsequently treated with afatinib starting in November 2020, and computed tomography (CT) imaging showed a partial response for the liver metastases (tumors 1 and 2 in Figure 3) until August 2021, when one liver tumor enlarged (tumor 2 in Figure 3). Tissue biopsy was performed on the progressive liver tumor, and the NGS panel ACTLung™ (ACT Genomics), which simultaneously evaluates 13 actionable genes and 8 fusion gene transcripts in cancer specimens, revealed a positive exon 20 p. T790M *EGFR* mutation in addition to the previously detected T751_I759delinsN *EGFR* mutation (Figure 4). The patient subsequently received osimertinib starting in November 2021, and CT imaging showed a mixed response for the liver tumors. A partial response was observed for tumor 2, but tumor 3 showed progression in January 2022 (Figure 5). Therefore, TS-1 treatment was suggested. Unfortunately, the tumors did not respond to subsequent treatment, and the patient died in May 2022.

In our case, after the detection of the T751_I759delinsN *EGFR* mutation by NGS, switching from treatment with TS-1 to treatment with afatinib achieved good responses in the adenocarcinoma metastases. Progression-free survival was 10 months for afatinib and 3 months for osimertinib in our study. NGS was effective for the identification of rare mutations and provided us with accurate information for selecting appropriate treatment, whereas PCR-based methods failed to detect the mutation. 

## 3. Discussion

We report the case of a patient with NSCLC who harbored an *EGFR* T751_I759delinsN mutation that was detected by NGS and developed an acquired T790M mutation. The T751_I759delinsN *EGFR* mutation was not detected by traditional PCR-based methods during the initial diagnosis or the detection of metastatic brain tumors. Lung adenocarcinoma is enriched in non-smoking women among east Asian populations, and most cases are driven by gene mutations, particularly *EGFR* mutations [10,11]. Because this patient had no smoking history, the probability that this lung cancer was associated with an underlying driver mutation was quite high. Therefore, we suggested the application of NGS to detect rare gene mutations. Using NGS, a driver gene mutation was eventually identified, and targeted therapy was prescribed, resulting in favorable progression-free survival. This outcome suggests that the routine application of highly sensitive methods is feasible for the detection of rare *EGFR* mutations, which are critical for predicting the response to EGFR-TKI therapy.

More than 10 druggable targets have been identified for NSCLC, and targeted therapies are greatly beneficial for patients with NSCLC associated with varying mutations and molecular characteristics [12]. However, without the correct detection of driver gene mutations, appropriate therapeutic decisions cannot be made, preventing the patient from receiving the most favorable treatments. In a previous report, an 11% false-negative rate for *EGFR* mutations was reported, with the most commonly missed mutations identified as exon 20 insertions (71%), followed by exon 19 deletions (29%). The missed mutations included A763_Y764insFQEA, N771_P772insN, V769_D770insGTV, and H773_V774insNPH in exon 20 and L747_A750 > P and T751_I759 > S in exon 19 [4,13,14,15]. Patients with these missed *EGFR* mutations may still benefit from the use of EGFR inhibitors. Therefore, an increasing need exists to perform more comprehensive genetic testing to ensure the identification of targetable candidates.

PCR-based methods fail to detect these rare mutations due to the limited coverage area provided by pre-designed primers. In PCR-based methods, DNA derived from FFPE tumor tissue obtained from NSCLC patients is denatured and annealed, resulting in the amplification of sequences that match the sequences of pre-designed primers for specific variants, leading to the selective amplification of those variants [16]. Therefore, PCR-based tests can only identify the specific variants that can be detected by the designed primers. The Scorpion ARMS and the cobas^®^ EGFR Mutation Test v2 are the most commonly used methods for detecting *EGFR* mutations in lung cancer in the clinical setting. The Scorpion ARMS IVD2 kit can detect 29 types of *EGFR* mutations, including 3 mutations in exon 18, 19 deletions in exon 19, 2 mutations in exon 20 (S768I and T790M), 3 insertions in exon 20, and 2 mutations in exon 21 (L858R and L861Q) [17]. ARMS can distinguish between a match and a mismatch at the 3′ end of a PCR primer [18]. By contrast, cobas v2 can identify 42 mutations in *EGFR* exons 18, 19, 20, and 21, including the T790M resistance mutation [19]. The primers in each kit are designed to detect specific mutations, and neither kit includes all possible mutation sequences, with neither kit including primers designed to detect low-frequency *EGFR* mutations. As a result, rare *EGFR* mutations are unlikely to be detected, which can result in a less-than-optimal therapeutic management for patients who harbor rare mutations.

Therapeutic decisions should be guided by a thorough understanding of the molecular features of tumor tissues. In this case, a T751_I759delinsN *EGFR* mutation was discovered using NGS, leading to a change in the therapeutic approach. T751_I759delinsN is an exon 19 mutation in *EGFR* associated with sensitivity to EGFR-TKIs, such as gefitinib, erlotinib, afatinib, and icotinib [20]. After switching TS-1 therapy to afatinib therapy, the tumors displayed a partial response that lasted 11 months. Before receiving targeted therapy, this patient had previously been administered multiple-line chemotherapy due to the false-negative mutation testing outcomes, and the chemotherapy was not expected to result in a good response. With the discovery of an *EGFR* mutation by NGS, the patient was able to receive more appropriate therapy, leading to better survival. However, resistance eventually developed after 11 months of afatinib treatment, consistent with reports from clinical trials [21,22] and previous studies [23,24].

Acquired resistance to EGFR-TKI therapies typically develops while the patient is receiving an EGFR-TKI and after a documented response or evidence of disease control [25]. Among resistance-inducing genes, the *EGFR* T790M is a rare mutation known to have an important clinical impact and has been identified as the most common mechanism for acquired resistance to EGFR-TKIs, observed in nearly two-thirds of cases that progress following successful EGFR-TKI therapy [26,27]. The poor therapeutic response to EGFR-TKIs is associated with the high allelic frequency of the T790M *EGFR* mutation [28,29,30]. In this report, the T790M mutation was detected by NGS, allowing us to understand the development of resistance to afatinib. Among T790M-positive patients, the third-generation EGFR-TKI osimertinib has been reported to provide a good effect [31].

## 4. Conclusions

In summary, we report a case of lung adenocarcinoma with a T751_I759delinsN mutation in *EGFR* exon 19 that was detected by NGS but not by traditional PCR-based methods. *EGFR* genotyping is fundamental for clinical decision-making; however, traditional PCR has a limited detection capability for rare genetic mutations due to its underlying mechanism, which can delay the application of appropriate treatment for patients with rare driver mutations. Therefore, NGS is an important strategy for treatment decision-making, providing a highly sensitive and quantitative gene mutation detection rate. Overall, our case demonstrates the importance and potential of NGS as a routine test for identifying rare gene mutations when selecting appropriate clinical treatment strategies for patients with lung adenocarcinoma.

## Figures and Tables

**Figure 1 ijms-23-13451-f001:**
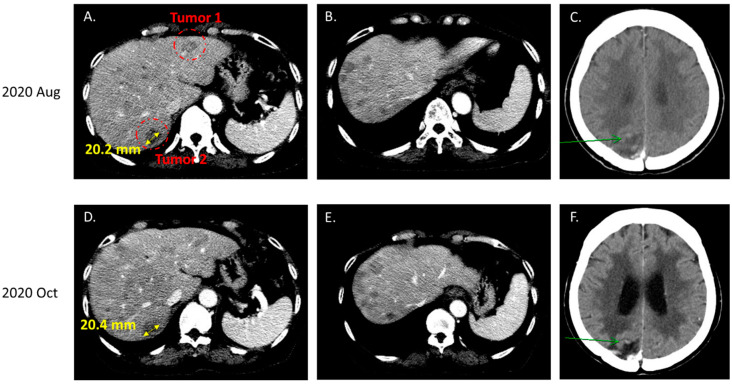
Computed tomography image showing stable metastatic tumors (**A**–**C**) before and (**D**–**F**) after TS-1 treatment. (**A**,**D**) Two measurable metastases (tumors 1 and 2) can be observed in the liver. Green arrows indicate metastatic tumor of brain.

**Figure 2 ijms-23-13451-f002:**
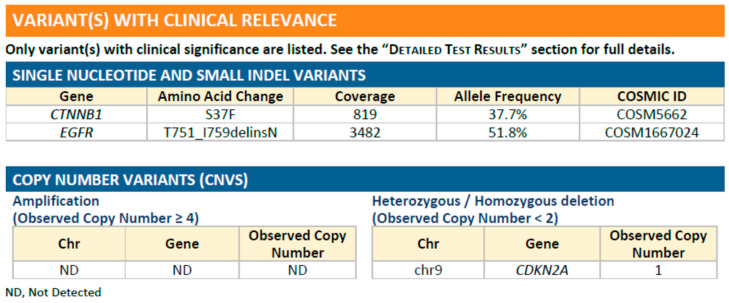
A targeted panel was used to simultaneously evaluate 40 actionable genes, 13 fusion genes, and more than 350 fusion transcripts (ACTDrug^®^+, ACT Genomics) associated with matched therapeutic options, which revealed a T751_I759delinsN mutation in EGFR.

**Figure 3 ijms-23-13451-f003:**
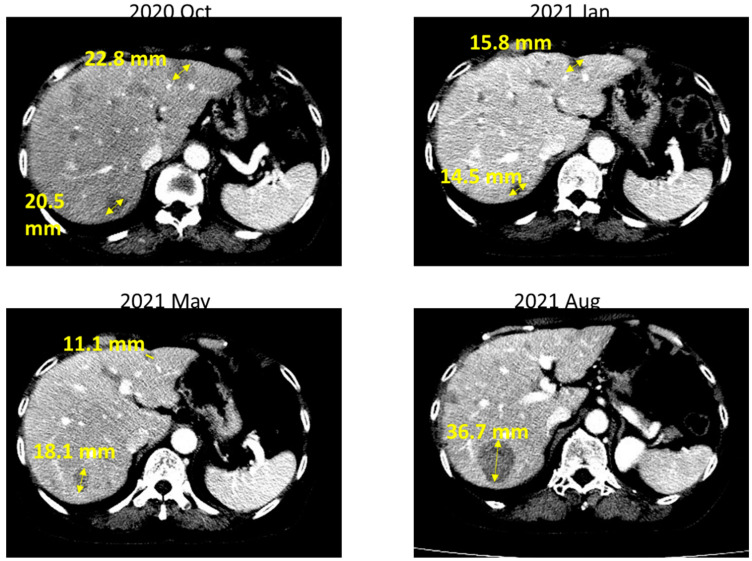
After afatinib treatment, computed tomography imaging showed a partial response for liver metastatic tumors 1 and 2. However, tumor 2 enlarged in August 2021, indicating progression.

**Figure 4 ijms-23-13451-f004:**
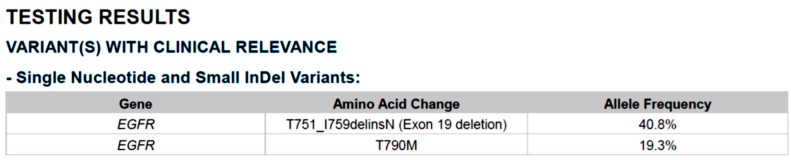
Tissue biopsy was performed for the progressive liver tumor, and an NGS panel that sequences 13 actionable genes and 8 fusion genes transcripts (ACTLung™, ACT Genomics) revealed a positive exon 20 p. T790M EGFR mutation in addition to the previously identified T751_I759delinsN EGFR mutation.

**Figure 5 ijms-23-13451-f005:**
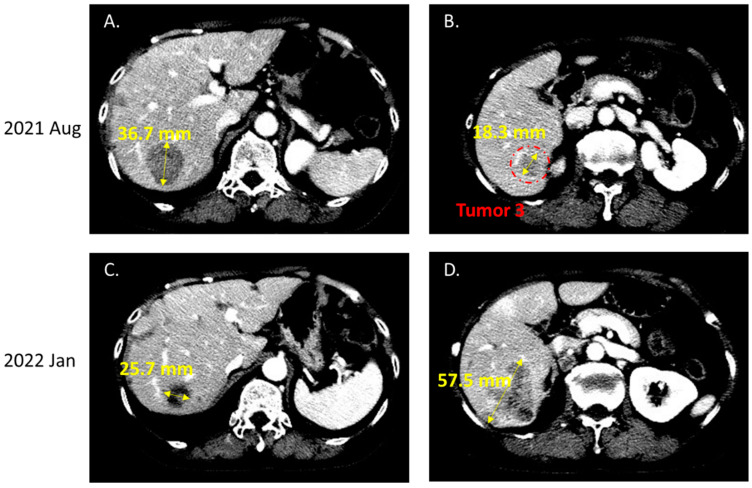
Computed tomography before (**A**,**B**) and after osimertinib treatment (**C**,**D**). After osimertinib treatment, computed tomography imaging revealed a partial response for liver tumor 2 (**A**,**C**); however, another tumor presented in the liver, which was identified as tumor 3, demonstrating progression (**B**,**D**).

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
