# Peer review of "EGFR T751_I759delinsN Mutation in Exon19 Detected by NGS but Not by Real-Time PCR in a Heavily-Treated Patient with NSCLC"

_ijms, 2022, doi:10.3390/ijms232113451_

Round 1

Reviewer 1 Report

The case report titled “EGFR T751_I759delinsN mutation in exon19 detected by NGS 2 but not by real-time PCR in a heavily-treated patient with NSCLC” reports the limitation of the current real-time PCR kits for detection of EGFR T751_I759delinsN mutation in exon19 in a heavily-treated patient with NSCLC. The rare mutation was finally identified by NGS. The similar case was reported as reference 7. Even though targeted therapy temporarily controlled the metastatic tumor progression, the patient still passed away. I assume the uncontrolled disease after application of osimertinib was due to secondary resistance. However, the info was not further provided. Overall, the case was clearly described.

Comments:

1.      The PCR method used for the patient should be included.

2.      As described in the second paragraph on page 2, there is similar rate for detection T790M mutation using NGS and digital droplet PCR. More than 60% of relapse to first and second generation of EGFR TKI treatment is due to T790M mutation. After relapse to the afatinib treatment, I wonder if NGS is necessary after relapse to afatinib treatment considering time and cost for the patient.

3.      Since Osimertinib showed efficacy superior to standard EGFR TKIs (1, 2), it may be a better choice for the patient. Even though secondary resistance to Osimertinib is still an unmet clinical problem.

4.       Whether the PCR kit could detect the rare mutation is largely limited by the primers pre-designed.

1.            Soria JC, Ohe Y, Vansteenkiste J, Reungwetwattana T, Chewaskulyong B, Lee KH, et al. Osimertinib in Untreated EGFR-Mutated Advanced Non-Small-Cell Lung Cancer. The New England journal of medicine. 2018;378(2):113-25.

2.            Ramalingam SS, Vansteenkiste J, Planchard D, Cho BC, Gray JE, Ohe Y, et al. Overall Survival with Osimertinib in Untreated, EGFR-Mutated Advanced NSCLC. The New England journal of medicine. 2020;382(1):41-50.

Reviewer 2 Report

The authors present a case report of a non-small cell lung cancer patient with a rare EGFR mutation identified on NGS when it was not found on standard testing with PCR.  They highlight prior studies of false negative tests using standard PCR testing and the role of NGS in identifying novel mutations.  The case report is concise, yet informative.  Overall, I feel it highlights an important clinical problem and cites the appropriate data.  
